# Changes in the Quality and Nontargeted Metabolites of Salt-Fermented Shrimp (*Saeu-jeot*) Based on Fermentation Time

Sunhyun Park [1,2,†], Keono Kim [3,†], Mi Jang [1], Heeyoung Lee [1], Jeehye Sung [3] and Jong-Chan Kim [1,*]

1. Food Standard Research Center, Korea Food Research Institute, 245 Nongsaengmyeong-ro, Wanju 55365, Republic of Korea; shpark@kfri.re.kr (S.P.); jangmi@kfri.re.kr (M.J.); hylee06@kfri.re.kr (H.L.)
2. Department of Food Science and Technology, Chungnam National University, 99 Daehak-ro, Yuseong-gu, Daejeon 34134, Republic of Korea
3. Department of Food Science and Biotechnology, Andong National University, 1375 Gyeongdong-ro, Andong 36729, Republic of Korea; rlarjsdh2803@gmail.com (K.K.); jeehye@andong.ac.kr (J.S.)
* Correspondence: jckim@kfri.re.kr; Tel.: +82-63-219-9155
† These authors contributed equally to this work.

**Abstract:** *Saeu-jeot* is a widely consumed variety of *jeotgal*, a South Korean salt-fermented food. However, there is a lack of existing studies conducting nontargeted metabolomic analyses of *saeu-jeot* during fermentation. To evaluate the changes in *saeu-jeot* during fermentation, *saeu-jeot* samples were fermented for 360 days under controlled conditions. Samples collected at different time points were subjected to physicochemical (including nontargeted metabolomic analysis) and microbial analyses. As fermentation progressed, the pH decreased and acidity increased, whereas total nitrogen, amino-nitrogen, and specific amino acid concentrations increased. Nontargeted metabolite analysis supports these results. Metabolite profiling classified changes in *saeu-jeot* during fermentation into those occurring in the early (15–45 days), middle (60–180 days), and late (270–360 days) stages. Pathogenic bacteria were not detected, and biogenic amine levels were not elevated, suggesting that *saeu-jeot* is safe to consume. Overall, pH, amino-nitrogen, and pathogenic bacteria, according to the fermentation stage of *saeu-jeot*, can be useful parameters for evaluating the quality of salted shrimp.

**Keywords:** *saeu-jeot*; salt-fermented shrimp; seafood fermentation; quality factor; quality changes; nontargeted metabolomics; traditional food

## 1. Introduction

Fermented products obtained by treating seafood with salt are traditionally consumed in Southeast Asia. Salt-fermentation technology helps store perishable raw food materials in warmer regions, such as Southeast Asia [1].

*Jeotgal*—a traditional salted and fermented seafood in South Korea—is prepared by adding 20–30% salt to 70–80 g of various marine organisms (e.g., shrimp, clams, oysters, and anchovies) without seed inoculation [2]. During fermentation and aging, free amino acids are released and low-molecular-weight peptides and aromatic components are produced through enzymatic reactions (such as protease activity), which impart *jeotgal* with a unique taste and flavor [3]. In South Korea, *jeotgal* is mainly used as a seasoning agent to add an umami flavor or as a sub-ingredient in foods such as *kimchi* [4,5]. *Saeu-jeot* and *myeolchi-jeot* (including *myeolchi-aekjeot*) are the most widely consumed varieties of *jeotgal* and are the main ingredients for *kimchi* fermentation [6].

Recent studies on the fermentation/aging process of Korean fermented seafood or fermented salted shrimp foods have focused on microbial growth characteristics. For example, Jung et al. [5] studied the effect of the fermentation period on changes in bacterial community composition in salted seafood. Differences in bacterial communities between commercial *saeu-jeot* and shrimp paste varieties have been studied [7–9], and strains isolated from *saeu-jeot* or *jeotgal* have also been characterized [10–12]. Helmi et al. [13] studied the

effect of the fermentation period on changes in the bacterial community composition and metabolite profile of *terasi*, a salted food from Indonesia similar to *saeu-jeot*.

Previous studies have used nontargeted metabolite screening to identify metabolic pathways [14,15]. Liquid chromatography coupled with high-resolution mass spectrometry (LC/HRMS) is used to determine metabolite profiles with high specificity and selectivity. Nontargeted screening is a promising approach for identifying metabolites from liquid chromatographic/mass spectrometric (LC/MS) data that provide in-depth metabolic information, including information about endogenous and exogenous chemical factors such as amino acids, nucleic acids, and organic acids [16]. Nontargeted metabolomic approaches are used for the global profiling of metabolomes to understand the characteristics of traditional fermented foods such as *kimchi*, *koji*, and *doenjang* [17–19]. However, LC/HRMS-based metabolic profiling of *saeu-jeot* during fermentation has not been explored to date.

In this study, the physicochemical and microbiological quality factors of Korean salt fermented shrimp was analyzed to evaluate the overall quality changes during the fermentation of salted shrimp. Furthermore, the metabolite profiles of salted shrimp samples collected during various fermentation periods were determined using LC/HRMS-based nontargeted screening methods. Accordingly, factors that can predict the quality and safety of salted shrimp were determined.

## 2. Materials and Methods

### 2.1. Sample Preparation

To analyze the effects of fermentation time on quality changes in *saeu-jeot*, samples at different fermentation/aging periods were prepared by Guldari Food Co. (Asan-si, Republic of Korea). Specifically, a mixture of tiny shrimp (*Acetes japonicus*) was mixed with sea salt purchased in Sinan-gun at a ratio of 75:25 ($w/w$) and fermented for 360 days in a room maintained at 10–15 °C and 85% humidity. Samples of this mixture taken on days 15 (15D), 30 (30D), 45 (45D), 60 (60D), 90 (90D), 120 (120D), 180 (180D), 270 (270D), and 360 (360D) of fermentation were transported to our laboratory within 2 h of the collection while maintaining the temperature at 4–10 °C.

### 2.2. Physicochemical Analysis

#### 2.2.1. pH and Acidity

To determine the pH and acidity, samples were collected by filtering only the liquid fraction through cotton gauze. pH was determined using a digital pH meter (Orion 3-Star Plus pH Meter; Thermo Scientific, Waltham, MA, USA). Acidity (expressed as a percentage of lactic acid) was determined via titration with 0.1 NaOH until the pH reached 8.3 [20].

#### 2.2.2. Total Nitrogen (TN), Amino-Nitrogen (AN), and Volatile Basic Nitrogen (VBN) Concentrations

TN concentration in samples was determined using the micro-Kjeldahl method [21]. AN was measured using the van Slyke method [22], and VBN was measured using a trace microdiffusion method on a Conway unit (Sibata Scientific Technologt, Co., Ltd., Tokyo, Janpan). TN and AN concentrations were determined using the liquid fraction (filtered through gauze, as mentioned above) of the samples, whereas the VBN concentration was determined using the solid from the whole sample.

#### 2.2.3. Biogenic Amine (BA) Concentration

The concentration of BAs in the sample was analyzed using the method proposed by Yoon et al. with slight modifications [23]. Briefly, the sample was homogenized by adding 25 mL of 0.1 N HCl to 5 g of the sample, centrifuging this mixture at $4000 \times g$ for 10 min at 4 °C, and filtering the supernatant through Whatman paper No. 4. A small amount (1 mL) of the sample and the mixed standard solution were added to a test tube with 0.5 mL of saturated $Na_2CO_3$ and 0.8 mL of 1% dansyl chloride solution and mixed; the mixture was then derivatized at 45 °C for 1 h. After shaking the mixture for 10 min

and subsequently adding 0.5 mL of 10% propylene solution and 5 mL of ether to it, the supernatant was collected, nitrogen-concentrated, and filtered with 500 μL of acetonitrile. A Capcell pak-C18 MG II (4.6 mm × 250 mm, 5 μm; Osaka-soda Co., Ltd., Osaka, Japan) column was used for HLPC analysis (Jasco, Tokyo, Japan). Mobile phase A was 0.1% formic acid in acetonitrile and B was 0.1% formic acid in water. The elution conditions of mobile phase A were as follows: initial stabilization at approximately 55%, 65% for 15 min, 80% for 20 min, and 95% for 40 min. The flow rate of the mobile phase was 1.0 mL/min, the sample injection amount was 20 μL, and the column temperature was 40 °C; the analysis was conducted at 254 nm. The standards, tryptamine, putrescine, cadaverine, histamine, serotonin, tyramine, spermidine, and dopamine hydrochloride, were purchased from Sigma-Aldrich (St. Louis, MO, USA). Spermine was purchased from Supelco (Bellefonte, PA, USA), while norepinephrine was obtained from Chemfaces (Wuhan, China). All reagents were of HPLC-grade quality.

### 2.3. Microbiological Analysis

To quantify bacterial populations in the fermented samples, 10 g of the samples was added to 90 mL of 0.1% buffered peptone water and homogenized for 60 s using an electrical homogenizer (400 Circulation; Seaward, London, UK), followed by 10-fold serial dilution. The diluted samples were then inoculated on a total viable cell count medium (aerobic count plate; 3M Microbiology Products, St. Paul, MN, USA), and cell counting was performed after 48 h of culture at 30 °C. To quantify yeast and mold, diluted samples were inoculated on yeast and fungi film (yeast and mold count plate; 3M Microbiology Products), and cell counting was performed after 48 h of culture at 30 °C. To quantify *Escherichia coli* and coliform bacteria, 1 mL of diluted sample was inoculated on the respective films (*E. coli*/coliform count plates; 3M Microbiology Products), and cell counting was performed after 48 h of culture at 30 °C. Among the red colonies observed, those surrounded by air bubbles were counted based on the dilution ratio to obtain the coliform count. Among the blue colonies, those surrounded by air bubbles were counted to determine the *E. coli* count.

Contamination with food-poisoning bacteria such as *Staphylococcus aureus* and *Vibrio parahaemolyticus* was also determined. *Staphylococcus aureus* was cultured on Baird-Parker agar plates (Difco BD, Franklin Lakes, NJ, USA) at 37 °C for 48 h. Black colonies surrounded by a transparent ring were isolated, inoculated on tryptic soy agar plates (Difco BD), cultured at 37 °C for 24 h, and subjected to 16S rRNA sequencing. Of these, only colonies identified as *S. aureus* were counted. *Vibrio parahaemolyticus* was cultured on thiosulfate citrate bile salt sucrose agar (Difco BD) at 35 °C for 24 h; turquoise colonies that were unable to ferment sucrose were counted.

### 2.4. Nontargeted Metabolite Profiling

2.4.1. Preparation of Metabolite Extracts

The whole sample of *saeu-jeot* was frozen in liquid nitrogen and ground to a fine powder using a mortar and pestle. The powdered samples (10 mg) were mixed with 1 mL of ACN/isopropanol/water (3:3:2, *v/v/v*) containing 2 μg/mL L-phenylalanine-1-$^{13}$C as an internal standard. The mixture was vigorously vortexed for 5 min and subjected to sonication in an ice-cooled ultrasonic bath (Branson Ultrasonics, Danbury, CT, USA) for 30 min. After centrifugation at 12,000× *g* for 20 min at 4 °C, the upper phase was collected, filtered through a 0.2 μM nylon membrane syringe (Whatman Inc., Sanford, ME, USA), and stored at −80 °C until LC-HRMS analysis. The pooled quality control (QC) sample was prepared by mixing an equal aliquot (20 μL) from all samples.

2.4.2. LC-QTOF-MS Analysis

We conducted a nontargeted metabolite analysis to analyze the metabolite profile of *saeu-jeot* during the fermentation period. The extraction of polar metabolites from *saeu-jeot* was optimized, and the mass information, including high-resolution-MS data, was obtained through nontargeted metabolite profiling using LC-QTOF/MS. To obtain MS/MS spectra

of the metabolites, analyses were performed in both MS/MS negative and MS/MS positive modes using a HILIC column.

Nontargeted metabolite profiling was conducted using an electrospray ionization quadrupole time-of-flight mass spectrometry (ESI-QTOF-MS) system (Agilent Technologies 6530, Santa Clara, CA, USA) coupled with an Agilent 1290 liquid chromatography system. The metabolite extracts of *saeu-jeot* were subjected to chromatography on an Agilent InfinityLab Poroshell 120 HILIC-Z column (2.1 mm × 150 mm, 2.7 μm) with an Agilent InfinityLab Poroshell 120 HILIC-Z guard column (2.1 mm × 5 mm, 2.7 μm) held at 30 °C. The binary solvent system comprised mobile phase A (10 mM ammonium acetate (pH 9.0) in water) and mobile phase B (10 mM ammonium acetate (pH 9.0) in 90% ACN) at a flow rate of 250 μL/min. The mobile phase comprised 10 mM ammonium acetate adjusted to pH 9.0 with ammonia. The gradient elution conditions were as follows: 90% B for 0–2 min, 90–50% B for 2–15 min, 50% B for 15–20 min, 50–90% B for 20–21 min, and 90% B for 21–30 min. The sample injection order was randomized to avoid any possible time-dependent changes during analysis, and pooled samples were injected every five samples for QC. The ESI-QTOF-MS was operated in the MS and auto MS/MS scan modes. The mass data were acquired in both negative and positive ionization modes at the mass range of 50–1700 $m/z$ with an isolation window of 1 $m/z$. The ESI parameters for the spectrometric detection were as follows: nitrogen was used as the drying gas at 300 °C at a gas flow rate of 10 L/min, the fragmentation voltage was optimized at 175 V, and product ion scan spectra were obtained at different collision energies (20 and 40 eV) in the auto MS/MS scan mode.

*2.5. Data Processing and Statistical Analysis*

Physicochemical and microbiological analyses and the BA experiments were performed in, at minimum, triplicate. All data are expressed as mean ± standard deviation. Statistical analysis was conducted using SPSS Statistics (version 20; IBM, Chicago, IL, USA). A one-way analysis of variance, using a randomized design and Scheffe's multiple range comparison test, was used to determine differences between samples. Differences were considered significant at $p < 0.05$ (using a 95% confidence limit).

The resulting mass chromatograms were analyzed using Agilent MassHunter Qualitative Analysis (version B.03.01). For metabolites detected using the LC-QTOF-MS-based metabolomics approach, the LC-MS raw files were converted into mzXML format using ProteoWizard's msConvert software. The extracted mzXML data files of the MS/MS chromatogram were then exported to XCMS Online "https://xcmsonline.scripps.edu (accessed on 1 October 2022)" for peak alignment, noise filtering, retention time correction, and peak area extraction [24]. The metabolite structure of potential biomarkers was tentatively annotated by searching the free databases of Scripps' Metlin (http://metlinscripps.edu/ (accessed on 25 September 2023)) using an accurate mass (<5 mg/kg) and MS/MS spectra matching. A principal component analysis (PCA) plot obtained from the XCMS Online standard output was used to visualize group clustering among the observations.

## 3. Results and Discussion

### 3.1. Changes in Quality Indicators during Saeu-jeot Fermentation

pH and acidity are indicators of the quality characteristics of fermented foods, including *jeotgal*, and the production of organic acids (such as lactic acid) by micro-organisms during the fermentation of *jeotgal* affects these quality factors [25]. Table 1 shows the changes in pH and acidity during the fermentation of *saeu-jeot*. The pH was 7.61 ± 0.03 on 15D, and it decreased to 6.98 ± 0.07 and 6.86 ± 0.05 at 60D and 360D, respectively. In general, salted seafood has a pH of 5.5 to 6.5, but crustacean salted seafood such as salted shrimp has a high pH due to the influence of amines [26]. The acidity level of 0.21% ± 0.02% at the early stage of fermentation (15D) was slightly increased to 0.61% ± 0.03% at a later stage of fermentation (120D). The changes in pH and acidity did not change significantly after 90D ($p < 0.05$).

**Table 1.** Physicochemical analysis of *saeu-jeot* samples at different fermentation time points.

| Fermentation Period (Days) | pH | Acidity (%) | Total Nitrogen (%) | Amino-Nitrogen (mg/100 g) | Volatile Basic Nitrogen (mg/%) |
|---|---|---|---|---|---|
| 15 | 7.61 ± 0.03 [a,*] | 0.21 ± 0.02 [a] | 1.18 ± 0.01 [a] | 381.84 ± 0.00 [a] | 9.73 ± 0.35 [a] |
| 30 | 7.28 ± 0.00 [b] | 0.38 ± 0.01 [b] | 1.06 ± 0.04 [b] | 449.83 ± 0.00 [a] | 13.37 ± 0.03 [b] |
| 45 | 7.14 ± 0.03 [bc] | 0.50 ± 0.05 [c] | 1.21 ± 0.01 [b] | 559.14 ± 3.47 [b] | 24.54 ± 0.05 [c] |
| 60 | 6.98 ± 0.07 [cd] | 0.58 ± 0.03 [cd] | 1.32 ± 0.00 [c] | 578.77 ± 0.00 [b] | 25.79 ± 0.08 [d] |
| 90 | 6.94 ± 0.04 [d] | 0.61 ± 0.03 [d] | 1.36 ± 0.01 [c] | 638.56 ± 3.45 [bc] | 26.83 ± 0.09 [e] |
| 120 | 6.80 ± 0.06 [d] | 0.66 ± 0.03 [d] | 1.42 ± 0.01 [d] | 661.58 ± 3.46 [cd] | 27.05 ± 0.01 [e] |
| 180 | 6.83 ± 0.01 [d] | 0.65 ± 0.00 [d] | 1.45 ± 0.00 [d] | 684.63 ± 2.12 [de] | 28.62 ± 0.13 [f] |
| 270 | 6.89 ± 0.10 [d] | 0.64 ± 0.01 [d] | 1.56 ± 0.00 [e] | 755.46 ± 8.87 [ef] | 40.78 ± 0.28 [g] |
| 360 | 6.86 ± 0.05 [d] | 0.66 ± 0.01 [d] | 1.59 ± 0.01 [e] | 777.87 ± 8.34 [f] | 39.38 ± 0.24 [h] |
| Change during fermentation ** | ↓ | ↑ | ↑ | ↑ | ↑ |

Values represent mean ± standard deviation. * [a–h] Different superscript letters within each column indicate significant differences ($p < 0.05$) between fermentation periods, based on Scheffe's multiple comparison test. ** ↓: decreases; ↑: increases.

The umami taste of fermented foods, such as *jeotgal*, is affected by nitrogen compound concentrations that vary throughout the fermentation and aging processes. Thus, TN, AN, and VBN concentrations were analyzed to determine quality changes in *saeu-jeot* samples at different fermentation stages. The concentrations of TN, AN, and VBN in the *saeu-jeot* samples that were allowed to ferment for 360 days increased over time (Table 1).

The TN concentration was 1.18% ± 0.01% at the early stage (15D), 1.32% ± 0.00% at the early–middle stage (60D), 1.42% ± 0.01% at the middle stage (120D), and 1.59% ± 0.01% at the late stage (360D) of fermentation, indicating a significant time-dependent increase ($p < 0.05$). Bekhit et al. [27] explained that the change in pH is due to the release of amino acids from proteins and polypeptides and the release of nitrogenous compounds due to microbial proteolytic activity. During fermentation, the protein of the raw material (shrimp flesh) of salted shrimp likely decomposed into nitrogen and escaped from the meat into the juice owing to osmotic pressure. The TN concentration is the most representative quality factor for fermented food. However, an increase in TN is a natural phenomenon in fermented-protein foods. Therefore, it is difficult to use it to confirm changes in quality according to the fermentation period, even though it can be used as a minimum criterion for checking the effectiveness of the fermentation process. Only products with a TN concentration of 1.0% and >0.8% for fish sauce and soy sauce, respectively, which are fermented foods, can be commercially sold in South Korea. The AN concentration in *saeu-jeot* was 381.8 ± 0.0 mg/100 g on 15D and 777.9 ± 11.81 mg/100 g on 360D. AN refers to the total amount of compounds with an amino group terminus in which nitrogen is present as a free amino group. AN, a chemical form of nitrogen in free amino acids, represents the total amount of compounds containing nitrogen groups and amino acids and is an important quality indicator related to the flavor and maturity of fermented foods, including fermented seafood [28]. Therefore, AN is used as an indicator of the degree of aging of fermented foods, including salted fish, and may be considered an important quality factor because it is strongly correlated with flavors such as umami.

The VBN concentration (a major indicator of the freshness of protein foods) was 9.73 ± 0.48 mg/% at the early stage (15D) of fermentation and 39.38 ± 0.34 mg/% at the last stage (360D) of fermentation. The VBN concentration is the amount of lower basic nitrogen compounds that are volatile (such as trimethylamine, ammonia, and various basic amines) and is a quality factor used to determine the freshness of marine products [29]. The VBN concentration was approximately 40 mg/% on 360D, suggesting a decrease in the freshness of samples over time. However, as the international standard value of VBN recommended for salt-fermented products is not well defined, this parameter cannot be used to reflect the freshness of the final product. In addition, the VBN is used as an

indicator of the fresh condition of living things rather than fermented products. Thus, further research is needed to determine the VBN concentration of properly ripened and over-ripe products by monitoring commercially available *saeu-jeot* products.

### 3.2. Changes in Safety Factors during Saeu-jeot Fermentation

We determined the bacterial populations in *saeu-jeot* according to the fermentation period. This study aimed to examine the overall chemical and microbiological characteristics of *saeu-jeot*. Specifically, we concentrated on quantitative evaluations of each microbial species to identify microbiological safety indicators. The bacterial count was $3.9 \pm 0.2$ log CFU/g in the early fermentation period (15D), and this value decreased over time, reaching $2.7 \pm 0.1$ log CFU/g after 270D. Coliforms, including *E. coli*, and fungal species were below the detection limit in all samples. The number of these microbes was as low as that of environmental bacteria (2–4 log CFU/g). The two major food-poisoning microbes, *S. aureus* and *V. parahaemolyticus*, were determined when the raw materials were contaminated before or during fermentation; however, even these were below the detection limit in samples collected throughout the fermentation period. Considering that no additional sterilization processes are conducted other than the cleaning of raw materials, maintaining minimum hygiene standards and high safety factors for *saeu-jeot* production is important to ensure that *saeu-jeot* can be consumed directly as a sauce without heating. Herein, microbiological contamination was low; however, in a previous study, contamination with *S. aureus* and *Bacillus cereus* was confirmed during the microbial safety monitoring of salted shrimp products [30]; in another study, the degree of contamination was different among products from different manufacturers [31]. This suggests that the careful management of raw materials can help control microbial contamination in fermented salted shrimp.

Table S1 shows the major metabolites, and Table S2 shows the results of the BA analysis (changes in concentration over time). The quantitative analysis of BA was additionally conducted as histamine and histidine were detected during the fermentation metabolite profiling experiment (Table S1). BAs are low-molecular-weight amine-containing compounds typically generated through the decarboxylation of amino acids by microbial enzymes during the fermentation of food [32]. Histamine, a type of BA, causes clinical illness when its concentration exceeds 100 mg% (100 mg free base per 100 g of fresh fish flesh) [33]. Histamine is produced by specific bacteria in fermented fish products through enzymatic histidine decarboxylation [34]. The metabolite profiling results revealed that histidine and histamine levels increased with fermentation time, but their levels were below the detection limit during real-time quantification. Among 10 types of BA, tryptamine was detected at the early (15D; 1.24 mg/kg) and late stages (180D; 1.33 mg/kg) of fermentation, but the levels were not significantly different (Table S2). As an excessive intake of BAs can adversely affect health by causing high blood pressure, headaches, dizziness, nervousness, and tachycardia, the concentration of BAs in fermented foods is an important factor in ensuring consumption safety [35]. However, we recorded low levels of BAs in *saeu-jeot*, ruling out any safety issues.

### 3.3. Changes in Metabolite Profiles during Saeu-jeot Fermentation

Based on the analysis of *saeu-jeot*, 155 metabolites with $p \leq 0.01$ were analyzed in the MS/MS negative mode, whereas 1873 metabolites with $p \leq 0.01$ were analyzed in the positive mode (Figure 1 and Figure S1). The PCA of the results obtained in the MS/MS negative mode and the MS/MS positive mode showed that the metabolite profiles of *saeu-jeot* were significantly differentiated at the early (15D and 45D), middle (60D–180D), and late (270D and 360D) stages (Figure 1). This tendency was similar to that shown by the physicochemical analysis results of AN, wherein the AN concentration exhibited significant differences at the early, middle, and late stages of fermentation. The *saeu-jeot* samples analyzed in the MS/MS negative mode were separated to the left, whereas those analyzed in the MS/MS positive mode were separated to the right, based on principal component 1 (PC1) with an increase in fermentation time. The first PC of *saeu-jeot* analyzed

in the negative mode showed a contribution rate of 31%, whereas the second PC showed a contribution rate of 18%, representing a total cumulative contribution rate of 48%. In contrast, the first PC of *saeu-jeot* analyzed in the positive mode showed a contribution rate of 31%, whereas the second PC showed a contribution rate of 13%, resulting in a total cumulative contribution rate of 44%.

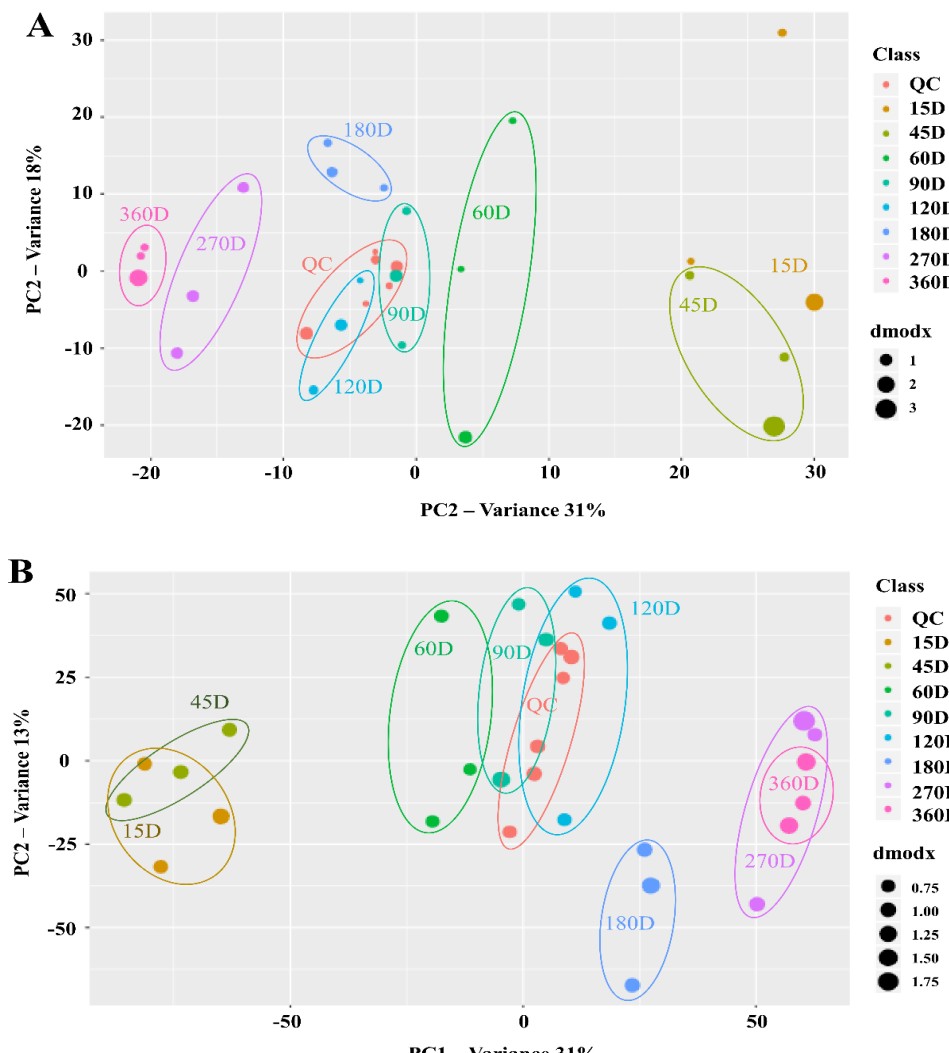

**Figure 1.** Principal component analysis (PCA) plot of chemical fingerprinting of *saeu-jeot* samples at different fermentation time points: (**A**) negative and (**B**) positive ionization modes. Samples collected on days 15 (15D), 30 (30D), 45 (45D), 60 (60D), 90 (90D), 120 (120D), 180 (180D), 270 (270D), and 360 (360D) were subjected to nontargeted metabolite profiling.

The major metabolites affecting the fermentation period of *saeu-jeot* were selected based on previously published studies [4,13] and confirmed with the MS/MS spectral tag library-based peak annotation procedure (Scripps' Metlin database (http://metlin. scripps.edu/ (accessed on 25 September 2023))) (Figure 2). The fermentation of *saeu-jeot* produces various metabolites, including organic acids and amino acids [1,5]. Among the organic acids identified in this study, fumaric acid showed a tendency to increase with a longer fermentation period. The acidity rapidly increased from 60D onward due to the production of fumaric acid and the increase in the concentration of several amino acids, such as aspartic acid and pyroglutamic acid. In addition, the concentration of amino acids such as tryptophan, pyroglutamic acid, histidine, tyrosine, aspartic acid, isoleucine, and asparagine increased rapidly from 60D onward. These amino acids are produced following the hydrolysis of shrimp proteins during fermentation and are important factors that impart

a unique taste to *saeu-jeot* [36,37]. Taste-active amino acids or their derivatives, such as aspartic acid and pyroglutamic acid, are the predominant tastants produced in various fermented foods and contribute to their particular umami taste [37].

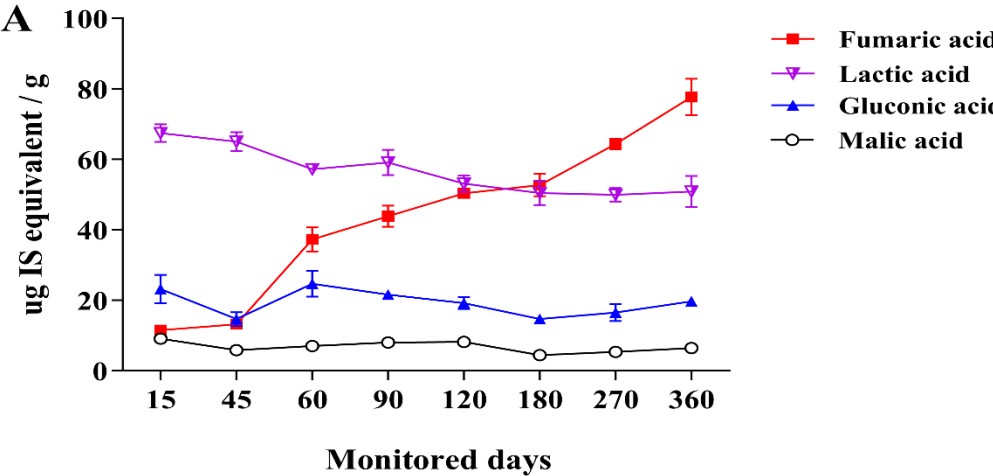

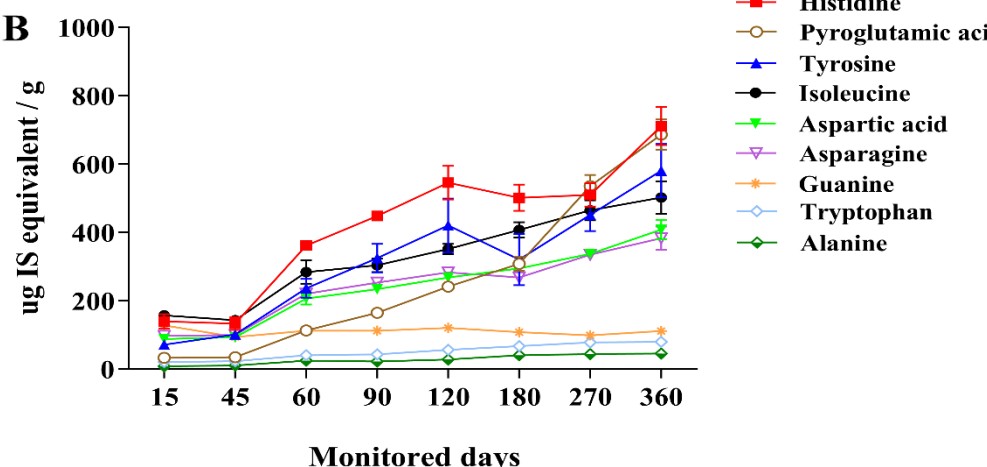

**Figure 2.** Concentration of major metabolites identified in *saeu-jeot* samples at different fermentation time points (bars represent standard deviation). IS, internal standard. (**A**) Changes in major organic acids and (**B**) Changes in major amino acids.

## 4. Conclusions

Using physicochemical and microbiological analyses and metabolite profiling, we assessed the overall changes in *saeu-jeot* during fermentation and aging processes. We also identified indicators for determining *saeu-jeot*'s quality. The pH of the *saeu-jeot* fermented for 1 year decreased, indicating weak basicity. Conversely, the acidity increased over time. Nontargeted metabolite profiling of samples from different fermentation and aging time points revealed that the increase in the concentration of organic acids (such as fumaric acid and lactic acid) is related to the fermentation time. A nitrogen-compound-related analysis showed that both TN and AN concentrations increased with the fermentation period, consistent with the results of metabolite profiling, which showed an increase in the concentration of amino acids. pH and acidity were correlated with an increase in organic acids, and changes in the concentration of AN showed a high correlation with changes in amino acids. This suggests that general component analyses such as pH, acidity, and AN can be used as quality factors to ensure the proper fermentation of salted shrimp. Furthermore, based on changes in each quality factor and metabolite profile, the

fermentation process can be statistically divided into early (15D–45D), middle (60D–180D), and late (270D–360D) stages. Product classification information based on the fermentation period (early, middle, and late stages) of *saeu-jeot* can help manufacturers adjust production timing to meet consumer preferences and/or demand. We did not find significant changes in the microbiological quality over the fermentation period, and pathogenic bacteria such as *S. aureus* were not detected. However, as *saeu-jeot* is consumed directly without heating, minimum microbial standards should be introduced by relevant food quality bodies. The quantitative analysis showed low levels, and therefore, low risk, of BAs in *saeu-jeot*. The levels of microbial contamination and BA production can be partially controlled through raw material management at the early stage of fermentation. Based on our results, pH (or acidity), AN, and pathogenic bacteria analyses are proposed to evaluate the quality of salted shrimp according to the fermentation period. The selected quality factors can be used as QC indicators in the production and manufacture of *saeu-jeot*. However, as our study could not confirm the quality level of all commercial fermented products, it is difficult to use our findings to set an upper or lower limit for the selected quality indicators. Based on the quality monitoring data of salted shrimp on the market, future studies should focus on determining the appropriate quality levels of fermented products.

**Supplementary Materials:** The following supporting information can be downloaded at https://www.mdpi.com/article/10.3390/fermentation9100889/s1: Figure S1: Cloud plot of chemical fingerprinting from multigroup analysis using XCMS Online software at different stages of *saeu-jeot* fermentation; Table S1: Potential selected and identified significant metabolites in *saeu-jeot* samples collected at different fermentation stages; Table S2: Change in BA concentrations in *saeu-jeot* samples collected at different fermentation stages.

**Author Contributions:** Conceptualization, S.P., K.K. and M.J.; methodology, S.P., M.J. and H.L.; validation, H.L.; formal analysis, K.K.; investigation, H.L. and J.S.; data curation, K.K.; writing—original draft preparation, S.P., K.K. and M.J.; writing—review and editing, S.P., J.S. and J.-C.K.; visualization, J.S.; supervision, J.-C.K.; project administration, J.-C.K. All authors have read and agreed to the published version of the manuscript.

**Funding:** This research was funded by the Korea Food Research Institute (funded by the Ministry of Science and ICT, Republic of Korea) (grant number E0211400-02) and the Basic Science Research Program through the National Research Foundation of Korea (funded by the Ministry of Education, Science and Technology, Republic of Korea) (project number 2020R1C1C1003766).

**Institutional Review Board Statement:** Not applicable.

**Informed Consent Statement:** Not applicable.

**Data Availability Statement:** Not applicable.

**Conflicts of Interest:** The authors declare no conflict of interest.

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
