# Peer review of "Changes in the Quality and Nontargeted Metabolites of Salt-Fermented Shrimp (Saeu-jeot) Based on Fermentation Time"

_fermentation, doi:10.3390/fermentation9100889_

Round 1

Reviewer 1 Report

1.The statistical method of difference used in the experiment should be more scientific.

2.How many biological and technical replicates were performed for each experiment? Please specify in the methods section as well as in the legend.

3.In the fermentation of seafood, microorganisms play a major role, among which there must be major microbial communities, which give the product excellent characteristics in the production process. Further metabolic studies were conducted on microbial communities, but the authors did not describe the microbial species category in the introduction section. Please explain the reasons and add relevant content in the Introduction section.

4.The characterization of product fermentation characteristics by metabolites and metabolic pathways is a relatively novel approach, but how to link metabolism with product changes, and whether some microbiome related life activities should be added to the discussion, please add to the discussion of metabolomics.

5.In this article, many sentences use the simple present tense, please check and revise carefully.

Major English revisions are required, and some of the issues are difficult for readers to understand.

Author Response

Thank you very much for taking the time to review our manuscript. Please find the detailed responses to each of your comments below. Additionally, changes made to the manuscript are indicated with tracks on in the re-submitted files. 

Comments 1: The statistical method of difference used in the experiment should be more scientific.

Response 1: In this study, all the analytical results (at minimum, triplicate) have been obtained through statistical analysis. Statistical analysis was conducted using SPSS Statistics software (version 20; IBM, Chicago, IL, USA). A one-way analysis of variance, using a randomized design and Scheffe’s multiple range comparison test, was used to determine differences between samples. Differences were considered significant at p < 0.05 (using a 95% confidence limit).

Comments 2: How many biological and technical replicates were performed for each experiment? Please specify in the methods section as well as in the legend.

Response 2: As you mentioned, it has been indicated in the methods section (line 187) that each experiment was conducted in triplicate for the purpose of statistical analysis.

Comments 3: In the fermentation of seafood, microorganisms play a major role, among which there must be major microbial communities, which give the product excellent characteristics in the production process. Further metabolic studies were conducted on microbial communities, but the authors did not describe the microbial species category in the introduction section. Please explain the reasons and add relevant content in the Introduction section.

Response 3: First, we would like to clarify that we did not conduct an overall microbial community analysis in this study. The analysis conducted in this study concerns non-targeted metabolites in saeu-jeot at different time stages during the fermentation process. Therefore, we have revised and incorporated this information in the manuscript as it may have been misunderstood (lines 263-267). Please take this into consideration. Our primary focus in this research was to examine the overall chemical and microbiological characteristics of saeu-jeot. Specifically, we concentrated on quantitative evaluations of each microbial species. Please understand that we did not submit information regarding microbial community data because the counts of various microorganisms analyzed through the agar plate method were all below the detection limit, except for the total bacterial count.

Comments 4: The characterization of product fermentation characteristics by metabolites and metabolic pathways is a relatively novel approach, but how to link metabolism with product changes, and whether some microbiome related life activities should be added to the discussion, please add to the discussion of metabolomics.

Response 4: In this study, we did not conduct an analysis of metabolic pathways, nor did we perform microbial community and microbiome analyses. Our focus was solely on the analysis of non-targeted metabolites, specifically organic acids and amino acids (including biogenic amines), to observe their variations across different fermentation periods, as revealed in previous studies (lines 329-330). We hope to explore the relationship between metabolites and the microbiome in our future work.

Comments 5: In this article, many sentences use the simple present tense, please check and revise carefully.

Response 5: Thank you for your guidance. We have thoroughly reviewed the entire manuscript and made appropriate corrections to the language and structure of the manuscript.

Response to Comments on the Quality of English Language:
We have sought the assistance of a professional English editing company to run language checks on our manuscript; however, in response to your feedback, we have requested another round of editing of our manuscript. We have also included the relevant English editing certificate. Thank you.

Reviewer 2 Report

Dear authors, the paper deals with a not so common type and lenght of fermentation process.

I have some questions for you:

- You did not wrote down any of the names of the microbial species isolated during the process. Why? Is it possible to include fungal and bacterial species names other than the two pathogenic bacteria already reported?

- Which technique was used to asses the microbial community composition? 

- If it is possible, make the colored lines in figure two thinner than those already reported. 

the authors at line 242 reported the following "We determined the bacterial community composition of saeu-jeot according to 242 fermentation and aging period (data is not shown)." My question was about this bacterial community: is it possible to know which technique was used to determine this community? Is it also possible to report any bacterial or fungal species, other from the pathogenic ones already mentioned?

Author Response

Thank you very much for taking the time to review our manuscript. Please find the detailed responses to each of your comments below. Additionally, changes made to the manuscript are indicated with tracks on in the re-submitted files. 

Comments 1: You did not wrote down any of the names of the microbial species isolated during the process. Why? Is it possible to include fungal and bacterial species names other than the two pathogenic bacteria already reported?

Comments 2: Which technique was used to asses the microbial community composition? 

Response 1&2: Thank you for pointing this out.

First, we would like to clarify that we did not conduct an overall microbial community analysis in this study. Therefore, we have included a clarification regarding this in the revised manuscript as it may have been misunderstood (lines 263-267). Please take this into consideration. Our primary focus in this research was to examine the overall chemical and microbiological characteristics of saeu-jeot. Specifically, we concentrated on quantitative evaluations of each microbial species to identify microbiological safety indicators. We conducted quantitative assessments for various microbial species, including analysis of known indicators of contamination such as E. coli and coliforms, to assess contamination of the saeu-jeot from the surrounding environment. Additionally, we determined the total bacterial count, yeast, and mold counts to establish basic information regarding the background bacterial population. Furthermore, we performed a quantitative analysis for two pathogenic bacteria.

The primary reason for selecting these two pathogenic foodborne bacteria is detailed below, and they were indeed detected in various previous studies on foodborne pathogens in similar foodstuffs (jeotgal).

- Staphylococcus aureus: Serves as an evaluation of hygiene quality among workers and in the working environment.

- Vibrio parahaemolyticus: Serves as an evaluation of hygiene quality in marine-derived raw materials.

Comments 3: If it is possible, make the colored lines in figure two thinner than those already reported. 

Response 3: Following your recommendations, we have adjusted Figure 2 to make the colored lines thinner than those already reported(lines 348-349). Also, we have replaced the images with higher-resolution versions, thereby meeting the journal's minimum recommended resolution of 600 dpi. Please review the updated attachments (lines 323-324, lines 348-349).

Reviewer 3 Report

This paper investigates the quality and nontargeted metabolomic analysis of Saeu-jeot. Although a great deal of work has been done by the authors, there are some implausible aspects of the present work.

1. The content of nitrite in the process of salt fermented foods is not only an indicator of quality characteristics but also of safety characteristics. The authors should add the determination of nitrite content throughout the process.

2. The images in Figure 1 are not clear, and authors are requested to show clear images in accordance with the submission requirements.

3. Microbiological analysis is used as an important indicator in fermented foods. The authors are requested to present in the text all the data in section 3.2 and to show pictures of the morphology of the relevant flora.

4. The sample of this experiment was selected from a single source, and in the Conclusion section the authors also elaborated that the results of this experimental study do not determine the level of quality of commercially available fermented products. The applicability of the author's conclusions is too low, is it still meaningful?

5. Please add different samples to refine the results of this experiment and improve the accuracy and applicability of the findings.

Author Response

Thank you very much for taking the time to review our manuscript. Please find the detailed responses to each of your comments below. Additionally, changes made to the manuscript are indicated with tracks on in the re-submitted files. 

Comments 1: The content of nitrite in the process of salt fermented foods is not only an indicator of quality characteristics but also of safety characteristics. The authors should add the determination of nitrite content throughout the process.

Response 1: Thank you for your valuable advice. As you pointed out, in naturally fermented foods, microbial activity can generate undesirable metabolites such as biogenic amines and nitrites. Moreover, there are concerns regarding the harmful effects of nitrites when used to maintain the reddish color of various processed foods. However, in the case of the saeu-jeot used in our study, no reports of nitrite detection have been documented; thus, we have refrained from analyzing this aspect in our study.

Comments 2: The images in Figure 1 are not clear, and authors are requested to show clear images in accordance with the submission requirements.

Response 2: As per your feedback, we have replaced the images with higher-resolution versions, thereby meeting the journal's minimum recommended resolution of 600 dpi. Please review the updated attachments(lines 323-324).

Comments 3: Microbiological analysis is used as an important indicator in fermented foods. The authors are requested to present in the text all the data in section 3.2 and to show pictures of the morphology of the relevant flora.

Response 3: First, we would like to clarify that we did not conduct an overall microbial community analysis in this study. We have included a clarification regarding this in the revised manuscript as it may have been misunderstood (lines 263-267). We conducted quantitative assessments for various microbial species, including the analysis of known indicators of contamination such as E. coli and coliforms to assess contamination from the surrounding environment. Additionally, we determined the total bacterial count, yeast, and mold counts to establish basic information about the background bacterial population. Furthermore, we performed a quantitative analysis for two pathogenic bacteria, namely, Staphylococcus aureus and Vibrio parahaemolyticus.

Please understand that we did not submit graphical data regarding microbial community structure because the counts of various microorganisms analyzed based on the agar plate method were all below the detection limit, except for the total bacterial count.

Comments 4: The sample of this experiment was selected from a single source, and in the Conclusion section the authors also elaborated that the results of this experimental study do not determine the level of quality of commercially available fermented products. The applicability of the author's conclusions is too low, is it still meaningful?

Comments 5: Please add different samples to refine the results of this experiment and improve the accuracy and applicability of the findings.

Responses 4 & 5: In this study, we produced saeu-jeot under restricted conditions to identify quality indicators. No previous study has tracked the quality changes in saeu-jeot over an extended fermentation period that lasted 360 days. We anticipated that by providing indicators for quality control (production process management), we could contribute to the management of saeu-jeot quality.

Furthermore, please understand that in this study, we have identified areas of deficiency (average and distribution analysis of values in commercially available products) and have suggested them as future research directions to promote advances in this field.